# Low Complexity Adaptive Detection of Short CPM Bursts for Internet of Things in 6G

**DOI:** 10.3390/s22218316

**Published:** 2022-10-29

**Authors:** Zihao Pan, Heng Wang, Bangning Zhang, Daoxing Guo

**Affiliations:** College of Communication Engineering, Army Engineering University of PLA, Nanjing 210001, China

**Keywords:** continuous-phase modulation, short bursts, soft input–soft output, adaptive algorithms

## Abstract

With the standardization and commercialization of 5G, research on 6G technology has begun. In this paper, a new low-complexity soft-input–soft-output (SISO) adaptive detection algorithm for short CPM bursts is proposed for low-power, massive Internet of Things (IoT) connectivity in 6G. First, a time-invariant trellis is constructed on the basis of truncation in order to reduce the number of states. Then, adaptive channel estimators, recursive least squares (RLS), or least mean squares (LMS), are assigned to each hypothetical sequence by using the recursive structure of the trellis, and per-survivor processing (PSP) is used to improve the quality of channel estimation and reduce the number of searching paths. Then, the RLS adaptive symbol detector (RLS-ASD) and LMS adaptive symbol detector (LMS-ASD) could be acquired. Compared to using a least-squares estimator, the RLS-ASD avoids matrix inversion for the computation of branch metrics, while the LMS-ASD further reduces the steps in the RLS-ASD at the cost of performance. Lastly, a soft information iteration process is used to further improve performance via turbo equalization. Simulation results and analysis show that the RLS-ASD improves performance by about 1 dB compared to the state-of-the-art approach in time-variant environments while keeping a similar complexity. In addition, the LMS-ASD could further significantly reduce complexity with a power loss of approximately 1 dB. Thus, a flexible choice of detectors can achieve a trade-off of performance and complexity.

## 1. Introduction

### 1.1. Background and Motivation

Globally, 5G networks have been deployed since 2020, and some key capacities are being standardized but do not meet the communication demands for 2030 and beyond. Researchers have begun to focus on 6G wireless communication networks to meet future requirements [1]. The IoT has recently drawn substantial interest due to its great potential for several applications in 6G wireless networks [2,3,4]. With the mass deployment of wireless sensors, it is difficult work to replace batteries or provide connections to the power grid due to random and complex environments. In these scenarios, the energy efficiency and cost of sensors are critical problems to be solved [5]. Future heterogeneous IoT integrated 6G networks would embrace active and passive devices in providing various services with diverse resources.

A backscatter communication technique [5,6,7,8] is proposed for passive devices that can harvest energy from the environment, providing the possibility to realize low power consumption and high-efficiency data transmission. In active devices with radio-frequency links, continuous phase modulation (CPM) is a good candidate for the communication of high power efficiency in uplinks [9,10,11]. Continuous phase modulation (CPM) provides a feasible transmission scheme for power- and bandwidth-limited wireless communication because of its high spectral efficiency and power efficiency [12]. Moreover, its constant complex envelope is robust to the nonlinearities caused by power amplifiers (PAs), which further improves power efficiency [13].

However, compared to linear modulations, CPM transmission over frequency-selective channels is a challenging task because of nonlinearity [11]. Poor physical-layer waveforms worsen the performance of the communication system, and even the superior power efficiency brought by PA can be undone because of the high complexity in the receiver [9]. Therefore, this paper focuses on the design of equalizers for IoT devices that employ low data rates and short CPM burst transmissions. In practical communications, the receiver does not have perfect channel state information, which must be estimated.

Optimal coherent detection can be approximately achieved by inserting a sufficient number of pilot symbols. In [14], a universal data-aided method was proposed by inserting periodically data-dependent symbols for CPM in flat Rayleigh fading channels. The channel response was generated by a mean squared error front-end filter and interpolation filter, which was input to a coherent CPM demodulator using the Viterbi algorithm. In fast-fading flat channels, the authors in [15] provided a data-aided channel estimation algorithm with local B-splines, and the results demonstrated an excellent trade-off between bit error rate performance and complexity. In [16], the authors derived a maximal posterior symbol detector for CPM over a double-selected channel via the PAM decomposition of the CPM. Data-aided initial channel estimation was achieved by using the pilot symbol, and the quality of the channel estimation was iteratively improved with turbo estimation. To further reduce complexity, a suboptimal equalizer based on a linear minimal mean-square error equalization was achieved at a cost of 2 dB. However, the data-aided method could drastically increase the overhead-to-payload ratio in short bursts because of additional training bits. Similarly, the addition of a cyclic prefix or unique words in low-complexity frequency-domain equalization (FDE) for CPM [17,18,19,20,21] could create the same problem. Thus, pilot- or FDE-based methods are not feasible schemes for short CPM bursts.

To solve this problem, blind channel equalization in the time domain is an effective alternative to avoid increasing the overhead-to-payload ratio. The authors in [22] used the Tong–Xu–Kailath algorithm to CPM and extracted the second-order statistics of the signal for channel estimation. The eigenvector method was used to identify the channel from a fourth-order cross-cumulant matrix under the GSM channel in [23]; to track the time-variant channel, the turbo-estimation approach was used to improve the quality of the channel estimates. However, the exact calculation of statistics requires a sufficient number of symbols to restrain the error due to noise, and the channel estimates obtained in short bursts are not reliable.

Other blind equalization can be achieved by the CPM–channel trellis. The joint trellis can be built from CPM and channel memory, which can be represented by a finite state machine (FSM). In [24,25,26,27], the FSM was described with a hidden Markov model (HMM). Thus, the channel response could be regarded as the unknown parameter in the HMM, which could be estimated with the expectation–maximization/Baum–Welch algorithm. However, HMM-based equalization techniques are suitable for time-variant channels. Thus, the idea that data-aided estimation could be embedded into the structure of the joint trellis [28,29]. A forward/backward adaptive soft-in soft-out (FAFB-SISO) algorithm [29] based on a full-state trellis was derived for linear modulation, which was optimal. Then, a forward adaptive SISO (FA-SISO) algorithm [30] that considered the channel correlation in only one direction was proposed for MSK; the author in [31] proposed various reduced-state A-SISO (RS-A-SISO) algorithms for reducing complexity.

Recently, a new noncoherent symbol-by-symbol detection algorithm of short quaternary CPM burst [9] was proposed as a better approach than the conventional one when short bursts are transmitted with only a few training bits. However, a major drawback of the detector is its high complexity because of matrix inversion for the calculation of branch metrics. In this paper, we propose a symbol-by-symbol detector with low complexity for short CPM bursts in IoT communication systems.

### 1.2. Contribution

The symbol detector is proposed on the basis of the trellis. To reduce the number of states and tackle a memory that linearly grows with time, a simplified trellis was constructed on the basis of the truncation of the memory.On the basis of the simplified trellis, the channel response is acquired by embedding the adaptive channel estimator, RLS or LMS algorithm, in the calculation of the trellis branch metric. The channel estimation is updated iteratively by using the recursiveness of the trellis, which can avoid matrix inversion when the LS estimator is used to calculate metrics and reduce complexity.In processing the trellis search, the BCJR algorithm was applied with per-survivor processing (PSP) in the forward direction, which could iteratively ensure the accuracy of channel estimation for each state.Lastly, a link-level simulation platform of the communication system was established to evaluate the performance of different symbol detectors.

### 1.3. Paper Outline

The rest of this paper is organized as follows. Section 2 provides the representation of CPM signals transmitted over fading channels and a communication system. Next, in Section 3, the new low-complexity symbol detector and the turbo scheme are introduced and are described in Section 4. The performance of the proposed symbol detector was evaluated and analyzed on the basis of the link-level simulation platform in Section 4. Lastly, the paper is concluded in Section 5.

## 2. System Model

### 2.1. CPM Representation

Before discussing the communication system model, we give the representation of a CPM signal. Let $a_{n}={\left[a_{0},a_{1},\ldots ,a_{n}\right]}^{T}$ be the vector containing independent and identically distributed (iid) symbols, where elements $a_{i}$ belong to the M-ary alphabet defined by $a_{i}\in \left\{\pm 1,\pm 3,\ldots ,\pm (M-1)\right\}$. The equivalent baseband representation $x(t,a_{n})$ of the transmitted M-ary CPM signal is written as follows:(1)$$x\left(t,a_{n}\right)=\sqrt{\frac{E_{s}}{T}}\exp\left(j\phi \left(t,a_{n}\right)\right),0\leq t\leq nT.$$
where *T* represents the symbol duration, and $E_{s}$ is the symbol energy. The information-bearing phase is defined as $\phi \left(t,a_{n}\right)=2\pi h\sum_{i=0}^{n}a_{i}q\left(t-iT\right)$, where *h* is the modulation index. $q\left(t\right)=\int_{0}^{t}g\left(\tau \right)d\tau$ is the phase pulse, such that q(t)=1/2 for $t\leq L_{cpm}T$, where $L_{cpm}$ is the memory length of the CPM. If $L_{cpm}=1$, we speak of a full-response CPM, and if $L_{cpm}\geq 1$, a partial-response CPM is obtained.

The CPM signal is transmitted through a fading channel, and distorted by additive white Gaussian noise (AWGN). In complex baseband representation, the received continuous-time signal is given by
(2)$$y\left(t\right)=x\left(t,a_{n}\right)*h\left(t\right)+v\left(t\right),0\leq t\leq nT.$$
where ∗ is the convolutional operator. A discrete-time representation of the received signal is needed for digital signal processing. Then, choosing a sampling period $T_{s}$, such that $T_{s}=T/\varepsilon$, where ε is the number of samples per symbol. We use notation $y_{k}=y\left(kT_{s}\right)$, $x_{k}=x\left(kT_{s}\right)$, $h_{k}=h\left(kT_{s}\right)$, $v_{k}=v\left(kT_{s}\right)$. Then, the *k*th sample of y(t) by fractionally spaced sampling every $T_{s}$ in Equation (Equation 2) is
(3)$$\begin{array}{cc} y_{k}=y\left(kT_{s}\right) & =\sum_{i=0}^{l-1}h_{i}\left(kT_{s}\right)x\left(kT_{s}-iTs\right)+v\left(kT_{s}\right)\\ \; & =\sum_{i=0}^{l}h_{i,k}x_{k-i}+v_{k}. \end{array}$$
where $T_{s}$ is the channel resolution, and *l* is the length of the channel in terms of $T_{s}$.

To obtain the matrix form for Equation (Equation 3), we give column vectors $x_{n}$, $y_{n}$, $v_{n}$ comprising the observables of $x(t,a_{n})$, y(t), v(t) from symbol time 0 up to time *n*. Therefore, the discrete-time representation of the received signal, CPM, and noise can be written as $y_{n}={\left[y_{0},y_{1},y_{2}\ldots ,y_{n\varepsilon -1}\right]}^{T}$, $x_{n}={\left[x_{0},x_{1},x_{2}\ldots ,x_{n\varepsilon -1}\right]}^{T}$, $v_{n}={\left[v_{0},v_{1},v_{2}\ldots ,v_{n\varepsilon -1}\right]}^{T}$. Thus, the *i*th path of the CPM signal can be defined as
(4)$$x_{n-i}={\left[0_{i},x_{0},x_{1},x_{2}\ldots ,x_{n\varepsilon -i-1}\right]}^{T}.$$
where $0_{i}$ is a row vector containing *i* zeros elements. All paths of the CPM signals were merged together and could be rewritten as
(5)$$X\left(a_{n}\right)={\left[x_{n},x_{n-1},\ldots ,x_{n-l+1}\right]}^{T}.$$

The channel response is $h={\left[h_{0},h_{1},\ldots ,h_{l-1}\right]}^{T}$ and the discrete-time representation of the received signal in the form of matrix can be written as
(6)$$y_{n}=X\left(a_{n}\right)h+v_{n}.$$

### 2.2. Serially Concatenated CPM System in Fading Channels

To derive the SISO symbol-by-symbol detector for CPM, we considered a serially concatenated CPM (SCCPM) setup for our system, as shown in Figure 1. When the CPM detector is SISO, CPM with channel coding could improve performance with a turbo equalization scheme in the form of SCCPM, where the decoder and detector exchange extrinsic information in terms of log-likelihood ratios (LLRs).

In the transmitter, a block of message bits m is encoded with a channel coder and permuted with an interleaver to randomize the order of the code bits. Subsequently, specific training bits are appended to the codeword in the training insertion block, helping the receiver in locking fast and reliably. An *M*-ary gray mapper was considered to obtain a symbol sequence by the obtained burst (in bits). Then, short CPM bursts are generated by a continuous phase encoder (CPE) and a memoryless modulator (MM) and transmitted over the channel. The channel was completely specified by linear distortions, represented by impulse response h(t,τ) and noise variance $\sigma_{v}^{2}$ that was assumed to be known at the detector.

On the receiver side, the discrete-time representation of the received signal is obtained by oversampling the output of the band selection filter. Then, given a priori LLRs $L_{A,CPM}$ (set zero at first iteration) from the decoder to detector, the latter uses the received signal and a priori symbol probabilities $P_{A,CPM}$ updated by $L_{A,CPM}$ to calculate the posterior probability $P_{CPM}$ of transmitted symbol $a_{n}$. Three steps are required from the $P_{CPM}$ to the a priori LLRs $L_{A,De}$, which is fed to the decoder.

First, the symbol probability is converted into codeword probability by the demapper, and then converted into the form of LLRs $L_{CPM}$. Subsequently, extrinsic information $L_{E,CPM}$ is computed by subtracting $L_{A,CPM}$, also called intrinsic information, from $L_{CPM}$. In turbo equalization, feeding back intrinsic information encounters problems of positive feedback, which leads to convergence to a suboptimal solution far from the globally optimal solution. Lastly, extrinsic information LLRs on the coded bits $L_{E,CPM}$ are deinterleaved to obtain a priori LLRs for decoder $L_{A,De}$. Just like subtracting intrinsic information, the interleaver is included in the iteration process to further disperse the direct feedback effect. These LLRs in the decoder are mapped again and similarly fed back to the detector. The process is repeated for a given maximal number of iterations or until convergence. Next, the detailed processing of the CPM detector is given.

## 3. Derivation of the SISO Symbol-by-Symbol Detector

### 3.1. Transition/Branch Metric

Bayes’ theorem shows that the posterior probability density function (pdf) of transmitted symbol vector $a_{n}$ is given by:(7)$$f\left(a_{n},h|y_{n}\right)=\frac{f\left(y_{n}|a_{n},h\right)f\left(a_{n}\right)f\left(h\right)}{f(y_{n})}.$$

The optimal estimated symbol sequence can be obtained by maximizing Equation (Equation 7). In the receiver, received signal $y_{n}$ is perfectly known and does not affect the maximization of Equation (Equation 7), which can be regarded to be a constant. Similarly, there is no prior information on the channel for the receiver, exerting no influence on the maximization problem. Therefore, the posterior pdf can be expressed as follows:(8)$$f\left(a_{n},h|y_{n}\right)∼f\left(y_{n}|a_{n},h\right)f\left(a_{n}\right),$$
where $f\left(a_{n}\right)=\prod_{i=0}^{N}a_{i}$ are the priori, which can be calculated with the extrinsic log-likelihood ratios (LLRs) from channel decoder. In standard equalization, i.e., with no iteration, $f(a_{n})$ are discrete uniform priors.

The conditional pdf of $y_{n}$ conditioned on the $a_{n}$, h is written as
(9)$$f\left(y_{n}|a_{n},h\right)=\frac{1}{{\left(\pi \sigma_{v}^{2}\right)}^{\varepsilon n}}exp(-\frac{1}{\sigma_{v}^{2}}\left|\right|y_{n}-X\left(a_{n}\right){h||}^{2}).$$

Logarithm likelihood function Equation (Equation 12) can be simplified by taking the logarithm, which is in the following form:(10)$$\Lambda_{n}\left(a_{n},h\right)=-\frac{1}{\sigma_{v}^{2}}\left|\right|y_{n}-X\left(a_{n}\right)h{\left|\right|}^{2}.$$

When the knowledge of the channel state information (CSI) is perfect, the most likely symbol sequence ${\hat{a}}_{ML}$ can be found with the maximization of Equation (Equation 10).
(11)$${\hat{a}}_{ML}=\underset{a\in A^{1\times n}}{\mathrm{argmax}}\Lambda_{n}\left(a_{n},h\right).$$

However, the channel response is unknown and can be estimated with the known transmission sequence denoted as $h_{est}=f\left[y_{n},a_{n}\right]$. Then, the branch metric of input symbol $a_{n}$ at the nth time slot is written as follows:(12)$$\gamma_{n}\left(a_{n}\right)=F\left[a_{n},y_{n},h_{est}\right].$$
where $h_{est}$ is the channel response estimated by the hypothetical transmit sequence and the received sequence before time *n*. Using the data-aided idea, suboptimal sequential maximal likelihood estimation can be achieved through a Viterbi-like algorithm [28]. However, in this paper, we focus on the derivation of the symbol-by-symbol detector rather than the sequence detector.

### 3.2. Derivation of SISO Symbol Detection

Given the branch metric, before deriving the symbol detector, an appropriate trellis needs to be constructed. On the one hand, when the metric depends on all the previous input symbols before the nth symbol, the memory grows linearly with sequence length [29], and a tree is built as shown in Figure 2, each node of which represents a sequence path. To construct a time-invariant trellis, the truncation of the tree memory can be used to force a fixed number of states in the trellis. On the other hand, when memory length *l* is constant, the complexity of the trellis search is proportional to number of branches $M^{l}$, which becomes intractable for large memory *l* and/or constellation size *M*. To reduce complexity, a simplified trellis with a reduced number of states can be built by truncating the memory. Therefore, the following derivation of the SISO symbol detector was implemented on the basis of the truncation of the trellis.

First, the symbol sequence can be divided into input symbols, trellis states at time *n*, and symbols obtained with PSP, which is defined as $a_{n}=\left[a_{n},\sigma_{n},P\left(\sigma_{n}\right)\right]$, where
(13)$$\sigma_{n}=\left[a_{n-1},a_{n-2},\ldots ,a_{n-L}\right],$$
is the state of the trellis after truncation, *L* is the truncation length, and $P(\sigma_{n})$ is the survivor sequence of arrival state $\sigma_{n}$ written as
(14)$$P\left(\sigma_{n}\right)=\left[a_{n-L-1},a_{n-L-2},\ldots ,a_{1},a_{0}\right].$$

Taking binary CPM with truncation length L=2 as an example, the truncated trellis is shown in Figure 2, where the state of trellis is $\sigma_{n}=\left[a_{n-1},a_{n-2}\right]$ and the branch metric defined by Equation (Equation 12). We used the BCJR algorithm to achieve symbol-by-symbol CPM detection. There were *M* branches from nth time slot to each state at (n+1)th time slot, and each branch was bundled with a channel response depending on all the symbols up to the nth symbol. Then, the appropriate channel response could be selected for the next state $\sigma_{n+1}$ by applying PSP in the forward recursion of BCJR, PSP-BCJR. The description of the PSP-BCJR algorithm for the symbol detection of CPM is given below.

In the forward recursion, channel estimation at states $\sigma_{n}$ can be defined as follows:(15)$$h\left(\sigma_{n}\right)=g\left(r_{n},\sigma_{n},P\left(\sigma_{n}\right)\right).$$

The channel response is calculated from the symbol sequence arriving at state $\sigma_{n}$, and the received sequence and g(·) imply the dependence of the channel response on the received and transmitted sequences, representing the estimation method. The state is transferred from $\sigma_{n}$ to $\sigma_{n+1}$ after entering symbol $a_{n}$. The computation of the branch metrics is determined with input symbol $a_{n}$, state $\sigma_{n}$ and its survivor sequences $P(\sigma_{n})$, and received signal $y_{n}$:(16)$$\gamma \left(\sigma_{n},\sigma_{n+1}\right)=F\left\{\sigma_{n}\to \sigma_{n+1},g\left[y_{n},a_{n},\sigma_{n},P\left(\sigma_{n}\right)\right]\right\}.$$

According to the branch metric, the forward recursion (initially zero) is written as follows:(17)$$\beta_{n+1}^{f}\left(\sigma_{n+1}\right)=\sum_{\sigma_{n}:\sigma_{n}\to \sigma_{n+1}}\beta_{n}^{f}\left(\sigma_{n}\right)\exp\left(\gamma \left(\sigma_{n},\sigma_{n+1}\right)\right)P_{n}\left(\sigma_{n+1}|\sigma_{n}\right).$$
where $P_{n}\left(\sigma_{n+1}|\sigma_{n}\right)$ is the probability that the state of termination $\sigma_{n+1}$ is transferred from $\sigma_{n}$, which is the prior probability of input symbols $a_{n}$. The prior probability was assumed to be uniform discrete values that could be updated by the extrinsic probabilities fed back from the decoder after every turbo iteration. As shown in Figure 3, PSP is used in forward recursion to simplify searching, and survivor sequences $P(\sigma_{n+1})$ and survivor channel estimation $h(\sigma_{n+1})$ of $\sigma_{n+1}$ can be selected according to
(18)$${\hat{\sigma }}_{n}=\underset{\sigma_{n}:\sigma_{n}\to \sigma_{n+1}}{\arg\max}\beta_{n}^{f}\left(\sigma_{n}\right)\exp\left(\gamma \left(\sigma_{n},\sigma_{n+1}\right)\right)P_{n}\left(\sigma_{n+1}|\sigma_{n}\right).$$

After completing the forward recursion, backward recursion is similarly calculated:(19)$$\beta_{n}^{b}\left(\sigma_{n}\right)=\sum_{\sigma_{n+1}:\sigma_{n}\to \sigma_{n+1}}\beta_{n+1}^{b}\left(\sigma_{n+1}\right)\exp\left(\gamma \left(\sigma_{n},\sigma_{n+1}\right)\right)P_{n}\left(\sigma_{n+1}|\sigma_{n}\right).$$

Lastly, a posteriori probabilities for symbol $a_{n}=u$ are the summation over all recursions with the same input *u*.
(20)$$P\left(a_{n}=u\right)=\sum_{u:\sigma_{n}\to \sigma_{n+1}}\beta_{n}^{f}\left(\sigma_{n}\right)\exp\left(\gamma \left(\sigma_{n},\sigma_{n+1}\right)\right)\beta_{n+1}^{b}\left(\sigma_{n+1}\right)P_{n}\left(\sigma_{n+1}|\sigma_{n}\right).$$
where $u:\sigma_{n}\to \sigma_{n+1}$ represent all transitions induced by input *u*. In practice, for feasibility and numerical stability, BCJR can be realized in the log domain called Log-MAP. Therefore, in subsequent simulations, the CPM detector was implemented in the log domain. More information and details can be found in [32]. Summarizing Equations (Equation 15)–(Equation 20), each branch is assigned a channel estimator at time *n*, which can track the variations of the channel while estimating the symbol by alternating between symbol and channel estimation. In [9], the unknown channel in Equation (Equation 13) is replaced with maximal likelihood (ML) estimation, and a suboptimal sequence metric is derived as follows.
(21)$$\begin{array}{cc} \Gamma_{ML}\left(a_{n}\right) & =\frac{1}{\sigma_{w}^{2}}y_{n}^{H}X\left(a_{n}\right)h_{ML}\left(a_{n}\right)\\ \; & =\frac{1}{\sigma_{v}^{2}}r_{n}^{H}X\left(a_{n}\right){\left(X^{H}\left(a_{n}\right)X\left(a_{n}\right)\right)}^{-1}X^{H}\left(a_{n}\right)y_{n}\\ \; & =\frac{1}{\sigma_{v}^{2}}y_{n}^{H}X\left(a_{n}\right)X^{+}\left(a_{n}\right)y_{n}. \end{array}$$

In order to derive the symbol detector, an incremental metric is used as the branch metric:(22)$$\Delta_{n}\left(a_{n}\right)=\Gamma_{n}\left(a_{n}\right)-\Gamma_{n-1}\left(a_{n-1}\right)=\frac{1}{\sigma^{2}}y_{n}X\left(a_{n}\right)h_{ML}\left(a_{n}\right)-\frac{1}{\sigma^{2}}y_{n-1}^{H}X\left(a_{n}\right)h_{ML}\left(a_{n-1}\right)$$
which is the implementation of F(·) in Equation (Equation 12). In fact, the authors in [9] assigned an ML estimator to each state that could be regarded as an instance of the symbol–symbol detector derived in this paper. Since the noise was white and Gaussian, the ML estimator was equivalent to the least-squares (LS) estimator, and the metric in [9] is noncoherent. Therefore, the detector is denoted as the least-squares noncoherent symbol detector (LS-NSD). However, a major drawback of the LS-NSD is its high complexity in the computation of branch metrics caused by matrix inversion in LS estimation. To simplify the calculation of the branch metric, some ingenious channel estimation methods can be considered to take place of the LS estimator.

In the process of trellis search, the update of channel estimation has recursiveness, and the forward recursion can be seen as a constant input of new data. It is easy to associate RLS and LMS [33] as a fast algorithm for LS estimation that can be used to process real-time data, which is highly convertible with the forward recursion in trellis search. To simplify the calculation of the branch metric, RLS or LMS can be applied in the channel estimation of each branch to avoid matrix inversion. In addition, both algorithms can identify the characteristics of the dynamic system in real time, allowing for the more accurate tracking of channel variations in a dynamic channel environment.

RLS requires the initial values of the channel response and the *P*, which can be initialized by the training sequence. However, under the constraint of blind equalization, it is assumed that the LS-NSD is still used to calculate the channel response for the first *t* time slots, and the RLS estimator is used after the (t+1)th time slots. The *P* is initialised by ${\left(X{\left(a_{t}\right)}^{H}X\left(a_{t}\right)\right)}^{-1}$ at the time *t*.

The gain vector at the kth moment (in terms of sample interval):(23)$$K_{k+1}=\frac{\frac{1}{\lambda }P_{k}x\left(t_{k}\right)}{1+\frac{1}{\lambda }x{\left(t_{k}\right)}^{T}P_{k}x\left(t_{k}\right)}.$$

Updating the inverse of the correlation matrix P at the (k+1)th moment:(24)$$P_{k+1}=\frac{1}{\lambda }P_{k}-\frac{1}{\lambda }K_{k+1}x{\left(t_{k}\right)}^{T}P_{k}.$$

Then, calculating the channel response:(25)$${\hat{h}}_{k+1}={\hat{h}}_{k}-K_{k+1}\left(x{\left(t_{k}\right)}^{T}{\hat{h}}_{k}-y_{k}\right).$$
where $x\left(t_{k}\right)={\left\{x_{k-i}\right\}}_{i=0}^{l-1}$ are fractionally spaced signal samples during channel length *l*. To further reduce complexity, the channel estimator for the branch metric can use the LMS algorithm:(26)$${\hat{h}}_{k+1}={\hat{h}}_{k}-\beta \left(x{\left(t_{k}\right)}^{T}{\hat{h}}_{k}-y_{k}\right)x^{*}\left(t_{k}\right).$$
where β is the step size chosen to be a suitable constant in LMS. To ensure the convergence of the algorithm, the range of the step satisfies
(27)$$0<\beta <\frac{\kappa }{tr[R]},0<\kappa <2,$$
where tr[R] is the trace of the autocorrelation matrix of $x(t_{k})$. In the case of a time-variant channel, normalized least mean squares (NLMS) can be used to improve tracking channel variations, and the step size is written as follows:(28)$$\beta \left(k\right)=\frac{a}{x{\left(t_{k}\right)}^{T}x\left(t_{k}\right)+\gamma },$$
where γ is a constant selected to prevent underflow.

Since the channel estimation methods are adaptive, the two symbol detectors proposed in this paper are denoted as the RLS adaptive symbol detector (RLS-ASD) and the LMS/NLMS adaptive symbol detector (LMS/NLMS-ASD).

### 3.3. Computational Complexity Analysis

Table 1 shows a detailed comparison of the complexity of RLS-ASD, LMS-ASD, and NLMS-ASD, and detection in [9] via floating-point operations (FLOP). The analysis of the computational effort of the different detectors is shown in Appendix A.

## 4. Simulation Results

### 4.1. Simulation Setup and Parameters

For the evaluation of the proposed blind turbo equalization, the parameters are summarized in Table 2, and the burst structure is shown in Figure 4. CPM with M=4, L=2−RC,h=1/2 was taken as an example for the experiment and simulation. CPM signals of each burst were sent over 4-tap equal power Rayleigh fading channels. In the first case, the single burst structure is shown in Figure 4a, with the training sequence using only a small number of bits to lock quickly and reliably for the receiver rather than for channel estimation in the conventional one. A time-invariant channel is usedthroughout a burst when the relative velocity of the transmitter and receiver is not very high, and the channel in any two different bursts is independent. In the second case, we considered a time-variant channel. The data bits were coded and divided into multiple subsequences as shown in Figure 4b, where each block of 20 code bits with a training sequence was transmitted as a short burst on a different channel.

### 4.2. Uncoded CPM: Single Burst and Time-Invariant Channel

In a single short burst, as shown in Figure 4a, the channel is assumed to be constant during a single burst and independent of each other. The uncoded performance of the different detectors in a time-invariant fading channel is shown in Figure 5. ‘Tran.’ curves represent the performance of the training-based method as the benchmark to beat. In a short burst, it is difficult for a few training data to provide an accurate estimation, resulting in poor performance. The LS-NSD curve is the performance of the symbol detector proposed as the baseline in [9], which served as the baseline of performance. In RLS-ASD, because of the time-invariant channel, RLS with forgetting factor λ=1 was applied to channel estimation in the branch metric. The performance of the RLS-ASD with 1 was roughly similar to that of LS-NSD, but with lower complexity.

Compared to LS-NSD and RLS-ASD, the LMS-based ASD had much lower complexity and practical feasibility, but its performance was strongly influenced by the step size. The LMS-ASD with fixed step sizes of κ=0.1,0.5,1.0,1.5,2.0 was selected according to Equation (Equation 27). BER curves show that the LMS-ASD with a greater κ achieved better BER performance than that of the smaller one in low $E_{b}/N_{0}$, but reached an error floor at high $E_{b}/N_{0}$. The results show that no error floor was observed for the LMS-ASD with κ=0.1, while there is a significant loss of performance at low $E_{b}/N_{0}$.

### 4.3. Coded CPM: Multiple Burst and Time-Variant Channel

At a low Doppler spread with $f_{d}T=0.002$, the BER performance of various symbol detectors for coded CPM is demonstrated in Figure 6. The performance of the four detectors without iteration is given in Figure 6a. The performance of the RLS-ASD with λ=0.99 was roughly similar to that of LS-NSD, but with lower complexity. BER curves show that the LMS-ASD achieved poorer BER performance than that of the NLMS-ASD since it utilized a fixed step size to achieve faster convergence, but ignored the steady-state misadjustment error. Therefore, NLMS-ASD had both faster convergence speed and a smaller misadjustment error. As the parameter *a* in Equation (Equation 28) was reduced, the performance of the NLS-ASD gradually improved, coming closer to that of RLS-ASD and LS-NSD.

After the third iteration, the comparison of the BER curves using various symbol detectors is shown in Figure 6. The curve of LS-ASD reached an error floor, while the BER of RLS-ASD had an obvious downward trend along with $E_{b}/N_{0}$. The performance of NLMS-ASD did not always increase with the decrease in parameter alpha, and the best performance was achieved with a=0.2.

The comparisons of LS-NSD, NLMS-ASD, and RLS-ASD are shown for the third iteration in Figure 7 for $f_{d}T=0.01$. At a high Doppler spread, LS-NSD could achieve slightly better performance than that of the one at a low Doppler spread since it obtained more gain from the diversity of the different channels with fast time variation. At the same time, as the Doppler spread increased, a smaller forgetting factor in RLS-ASD could achieve faster convergence. Then, NLMS-ASD with a=0.2 could achieve similar performance to that of LS-NSD. However, its complexity was lower than that of the LS-ASD or RLS-ASD. Thus, the NLMS-ASD with appropriate parameters could achieve an excellent trade-off between complexity and performance.

Lastly, we consider the performance of various symbol detectors when different channel codes and interleavers are employed. The BER after three iterations at $f_{d}T=0.01$ for the coded modulation schemes given in Table 2 and Table 3 is shown in Figure 8. Compared with the benchmark of Scheme A, the performance could be improved with the convolutional codes with a larger number of states in Scheme B, especially for RLS-ASD. However, the longer interleaving length in Scheme C did not significantly improve performance, which may have been due to the smaller frame length of the signal.

## 5. Conclusions

The paper proposed a class of symbol-by-symbol detectors based on the BCJR algorithm. The channel estimator was embedded in the forward recursion of the BCJR algorithm, and we illustrated the LS-NSD in [9] as an instance of the proposed detector. First, the performance of RLS-ASD and LMS -ASD with different step sizes was compared for a single CPM burst without coding and a time-invariant channel, using LS-NSD as a benchmark. RLS-ASD with λ=1 achieved similar performance to that of LS-NSD because RLS is a fast implementation of LS. LMS-ASD approached the benchmark as the step size decreased while reducing the FLOP number by about half.

In the second case, we considered multiple bursts with time-variant channels in the form of SCCPM. LS-NSD showed an error floor with different degrees at different maximal normalized Doppler spreads, while RLS-ASD achieved a performance improvement of 1 dB, which showed no or a light error floor. NLMS-ASD with appropriate parameters reduced the required number of FLOP by 45% at the expense of 1 dB power. Lastly, the performance of different code and interleave schemes are observed. Convolutional codes with a larger number of states could achieve more performance gains, most notably with RLS-ASD, while a larger interleaving depth could reduce performance.

For the short-burst CPM, in the case of time-invariant channels, the choice of different detectors could achieve a flexible trade-off between complexity and performance when the LS-NSD was a benchmark. In the case of time-variant channels, the proposed detector was more robust to time-variant environments than LS-NSD was. Furthermore, the limited computing capacity in IoT nodes motivates exploring other methods for reduced-complexity detection, which further reduces the amount of calculation. In addition, research on the influence of synchronization on our proposed method is practically significant, since perfect synchronization was assumed in the front end of our article. The mentioned open points are left for future research on the CPM receiver.

Moreover, in the future, 6G wireless networks would not be limited to only terrestrial communication, but would also be supplemented by nonterrestrial communication, especially the application of UAVs. In UAV-aided IoT communication systems, UAVs can provide more opportunities for line-of-sight paths and improved coverage range due to their flexible deployment and controlled mobility. In addition, a physical-layer innovation technique, intelligent reflective surfaces (IRS) [34,35,36], can reconfigure a wireless propagation environment by intelligently reflecting signals from the transmitter to the receiver, which can improve the performance of networks but remains unexplored. The design of the receiver with the assistance of UAVs and IRSs is a natural future research topic. The mentioned open points are left for future research on the CPM receiver.

As future work, there are three prospective directions: (1) searching for another method of reduced-complexity detection to further achieve a balance between complexity and performance; (2) studying the synchronization techniques of CPM, and analyzing the interplay between detection and synchronization; (3) exploring the design of CPM receivers with the assistance of UAVs and IRSs.

## Figures and Tables

**Figure 1 sensors-22-08316-f001:**
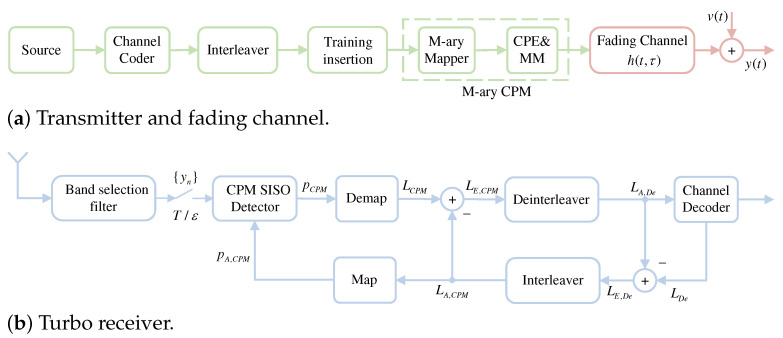
Communication system model of CPM, including transmitter, channel, and turbo receiver.

**Figure 2 sensors-22-08316-f002:**
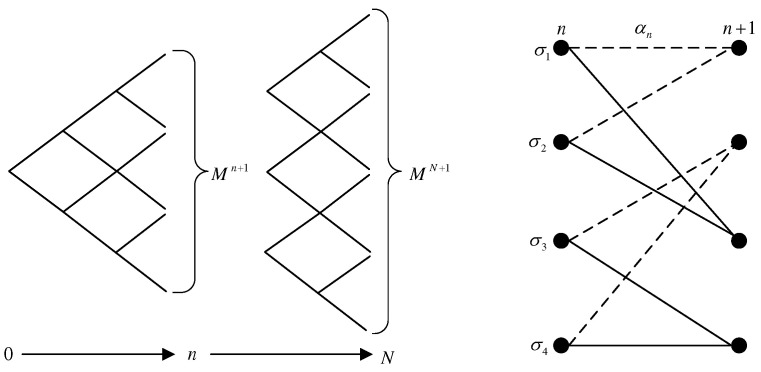
(**left**) Forward tree structure; (**right**) trellis diagram of the binary scheme with truncated length L=2 at time *n*.

**Figure 3 sensors-22-08316-f003:**
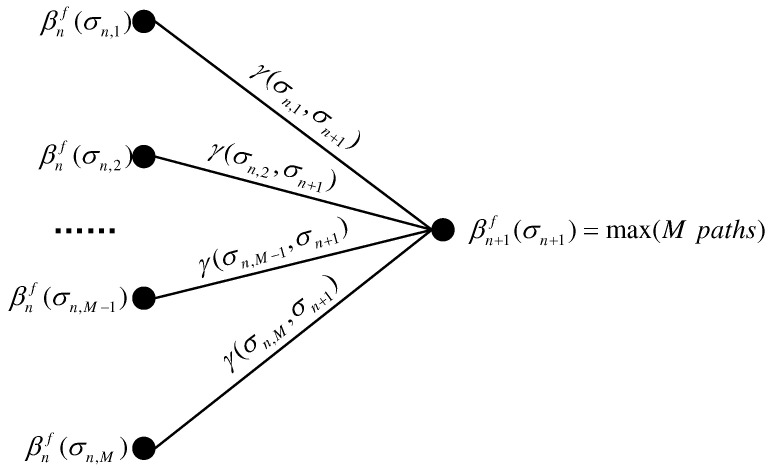
PSP-based forward recursion in trellis.

**Figure 4 sensors-22-08316-f004:**
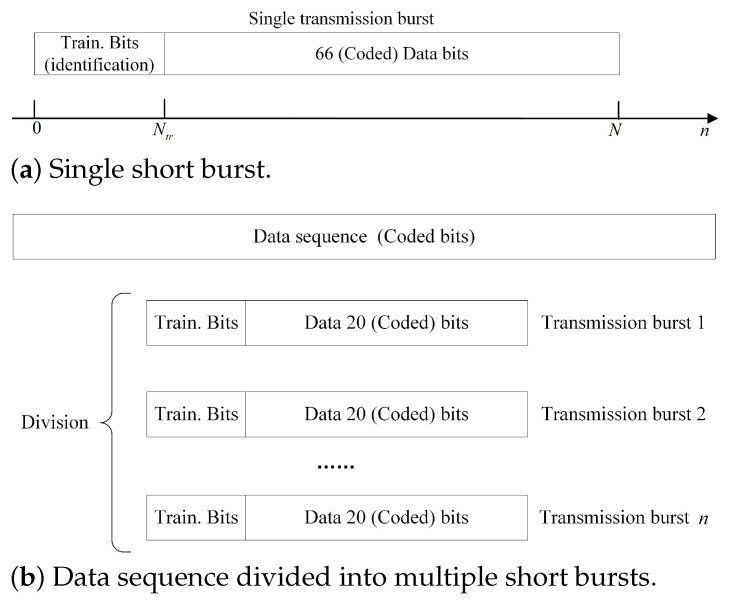
Short burst structure. (**a**) Single transmission burst at the output of the training insertion (TI) block with 66 payload coded bits. (**b**) Data sequence (in bits) is divided into *n* sequences of 20 bits, where each sequence was transmitted as a burst on a different channel.

**Figure 5 sensors-22-08316-f005:**
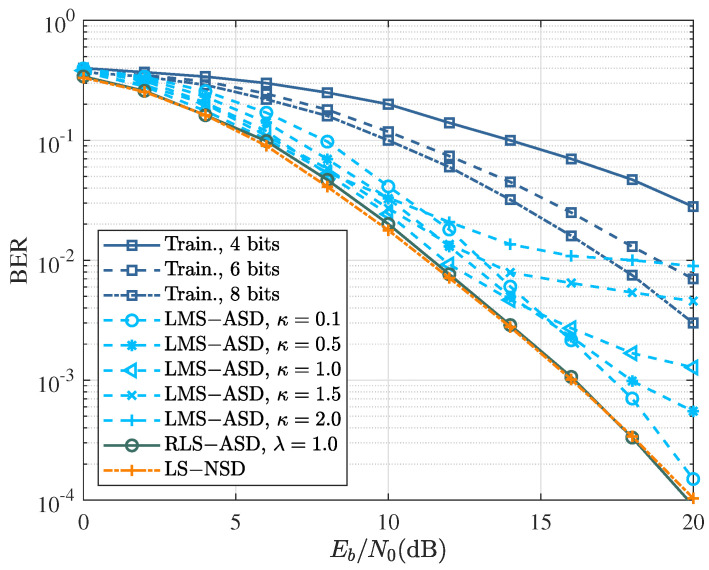
BER as a function of $E_{b}/N_{0}$ for different detectors. Curves Tran. and LS-NSD refer to the performance of training-based channel estimation and the symbol detection proposed in [9], respectively, representing the benchmark to beat.

**Figure 6 sensors-22-08316-f006:**
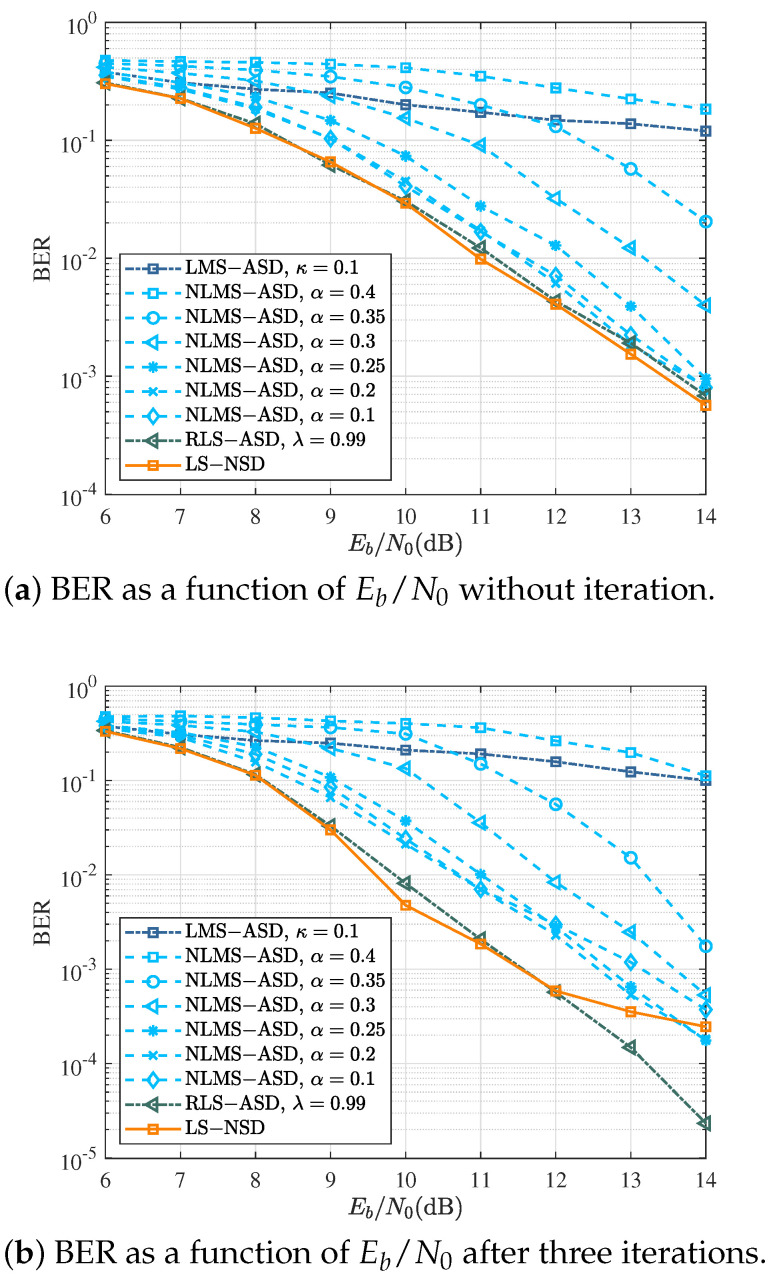
BER performance of various turbo symbol detectors at $f_{d}T=0.002$ for Scheme A in Table 3.

**Figure 7 sensors-22-08316-f007:**
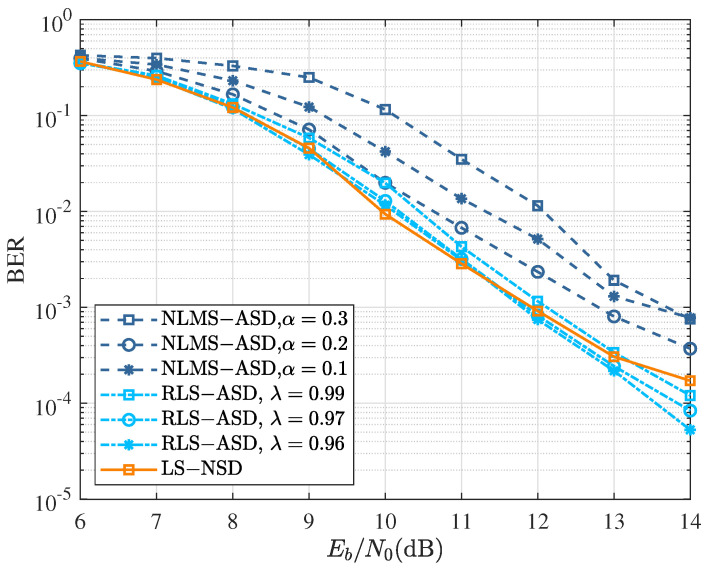
BER performance of various turbo symbol detectors after three iterations at $f_{d}T=0.01$ for Scheme A in Table 3.

**Figure 8 sensors-22-08316-f008:**
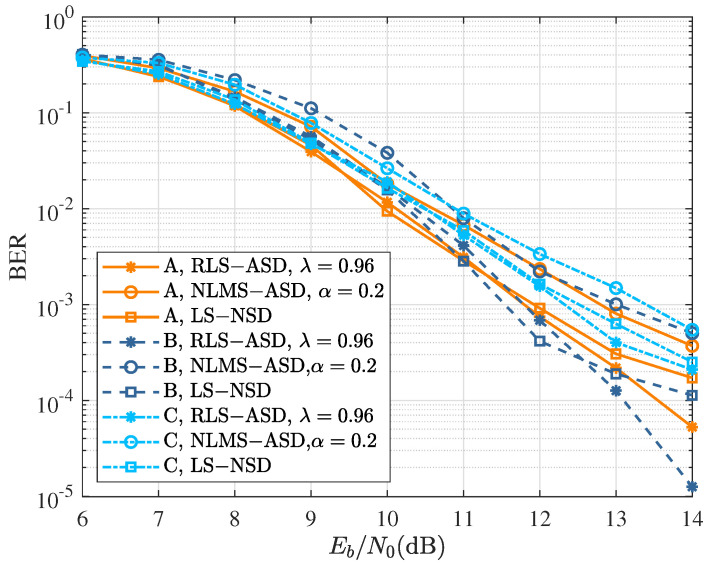
BER performance of various turbo symbol detectors after three iterations at $f_{d}T=0.01$ for the coded modulation schemes given in Table 2 and Table 3.

**Table 1 sensors-22-08316-t001:** Complexity comparison among detectors.

Detector Type	FLOP at *n*th Transition Interval
LS-NSD	$l^{3}+2l^{2}\varepsilon +2l\varepsilon +2\varepsilon nl+l+1$
RLS-ASD	$6l^{2}\varepsilon +2l\varepsilon +2\varepsilon nl-\varepsilon$
LMS-ASD	2nlε+3εl
NLMS-ASD	2nlε+5εl−ε

**Table 2 sensors-22-08316-t002:** Basic simulation parameters.

Parameters	Value	Remarks
Frequency pulse	Raised-cosine pulse	L=2
Modulation order (M)	4	-
Modulation index (h)	1/2	h=p/q
Mapping	Gray	-
Training bits	4,6,8	ML estimation
Coderate	1/2	Convolutional code
Samples/Symbol	2	ε
Multipath channel	Four-tap Rayleigh fading channels	Equal power

**Table 3 sensors-22-08316-t003:** Parameter of the coded and interleaver schemes; see also Table 2.

Scheme	Outer Code	Interleaver Length	Iterations
A	$R_{c}=1/2,{\left[5,7\right]}_{8}$	4831	3
B	$R_{c}=1/2,{\left[17,15\right]}_{8}$	4831	3
C	$R_{c}=1/2,{\left[5,7\right]}_{8}$	27,221	3

## Data Availability

Not applicable.

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
