# Peer review of "Low Complexity Adaptive Detection of Short CPM Bursts for Internet of Things in 6G"

_sensors, 2022, doi:10.3390/s22218316_

Round 1

Reviewer 1 Report

In this paper, a new soft-input-soft-output (SISO) adaptive detection algorithm for short CPM bursts is proposed for low-power, massive IoT connectivity in 6G. First, a time-invariant trellis is constructed based on truncation in order to reduce the number of states. Second, adaptive channel estimators, recursive least squares (RLS) or least mean squares (LMS), are assigned to each hypothetical sequence by using the recursive structure of the trellis, and per-survivor processing (PSP) is used to improve the quality of channel estimation and reduce the number of searching paths. The choice of different estimators allows a flexible trade-off between complexity and performance. Simulation results show the excellent trade-off between complexity and performance for the proposed symbol detector compared to the state-of-the-art approach. Overall, this work is written and organized well. However, I have the following comments:

1) Please pronounce CPM in the Abstract as "Continuous phase modulation (CPM)" In addition, define all the abbreviations in the first place.

2) Figure 1 should be improved as it is not visible. I think it can be designed in a better way. 

3) The related work can be further improved. In this version, the authors report only a few works; thus, this paper's novelty is doubted.

4) I suggest authors search all related papers and report them in the revised version. A few papers that might be very interesting are: Towards intelligent IoT networks: Reinforcement learning for reliable backscatter communications; Secure backscatter communications in multi-cell NOMA networks: Enabling link security for massive IoT networks.

5) It is better to make a table for notations and abbreviations used in this paper. It would make it easy for the reader of this paper.

6) In Section 2, the authors have started a subsection directly without saying anything. Please write a few sentences before the subsection which discuss the purpose of those subsections.

7) This work failed to report any work from 2021 and 2022, which is important to show how timely this work is. 

8) Some recent works on IoT include, Backscatter sensors communication for 6G low-powered NOMA-enabled IoT networks under imperfect SIC; NOMA-enabled backscatter communications for green transportation in automotive-Industry 5.0.

9) Study more recent works and report in this paper.

10) Can you justify the simulation parameters?

11) Is it possible to compare your work with the literature?

12) Extensive proofreading is suggested to improve the reading of this paper.

Reviewer 2 Report

File attached

Reviewer 3 Report

1. The authors claimed that 5G has been deployed everywhere, but it is still in its initial phase of deployment in many countries. Even the deployment of IoT is not up to industrial or home requirements. 

2. A better background is required to justify the claim. Suggest adding a few works of literature on it for better soundness of the introduction that will give a good flow of the paper. 

3. In line no 67, MSK technology has been mentioned, but how MSK fulfills the IoT-based application is not clear.

4. Diagram 1, needs to be more clear with better quality

5. Fig 6, it is simulated work (than which platform) else it is other?

6. References are not adeuete. 

Round 2

Reviewer 1 Report

Thank you so much for addressing my comments. The paper looks improved in this version. I have some minor comments. For example, I suggest the authors proofread the article for grammar errors and typos. Moreover, the conclusion can be revised so that potential future research directions must be added. Further, opportunities for physical layer security in UAV communication enhanced with intelligent reflective surfaces should be studied and reported.

Reviewer 2 Report

The manuscript is improved from its last version. But there are still some concerns that need to be addressed before final acceptance

1. The quality of Fig.1  should be improved further.

2. There are still typos and grammatical mistakes at certain points in the manuscript.

3. As pointed in the last review, the abstract and conclusion should be improved by numerically pointing how the RLS-ASD outperforms the state-of-the-art approach in time-variant environments. Should explain it with some numerical values (quantitatively) in terms of attained BER, performance improvement in percentage, reduction in computational complexity (by how much percentage.).

4. The future scope of the research study should be added in the conclusion.
